# Family and Youth Development: Some Concepts and Findings Linked to The Ecocultural and Acculturation Models †

**John W. Berry**

Department of Psychology, Queen's University, Kingston, ON K7L 3N6, Canada; elderberrys@gmail.com
† This paper is dedicated to the memory of my good friend and colleague James Georgas (1934–2018).

**Abstract:** Much research on migrants has focused on single individuals; however, the large-scale movement of people from one society to another often includes families made up of parents, their children and other relatives. Over time, these families and their members settle into their new society; they experience the process of acculturation and eventually adapt to their new circumstances. The processes of acculturation and adaptation are highly variable across cultural groups, societies of settlement, families and individuals. Sometimes this process is challenging, and may engender disagreements and conflicts among members of a family about how to acculturate. Variations in these patterns allow for the examination of which acculturation experiences and strategies lead to better adaptations. This paper reviews some of the core concepts and frameworks for examining them, and presents some findings on how families and youth acculturate and adapt. It concludes with some suggestions for how to acculturate using the integration strategy to improve family and individual adaptations.

**Keywords:** acculturation; acculturation strategies; adaptation; cultural transmission; ecocultural model; enculturation; ethnocultural groups; families; integration; youth



## 1. Introduction

As migration continues at increasing rates around the world, many disciplines have examined the demographic, economic, political, social and psychological phenomena that lead to, and result from, these migrations. Within this broad field of study, families are frequently ignored in psychological research on migration, with most of the focus being placed on individuals. However, much migration often takes place in family units, either at the time of first travel, or later under family reunification programmes. These units often include many members. such as children and grandparents [1]. This article reviews some conceptual frameworks that will allow us to focus our work on families, including parents and children. They provide guides for where to look when attempting to understand the migration and settlement of families and youth. It will also provide some examples of empirical research to illustrate this focus, and then consider some implications for future research and practice.

## 2. Materials and Methods

### 2.1. Ecocultural Approach

In our attempts to understand migration, it is important to know 'where people are coming from'. This requirement can be met by first noting that all human behaviour develops and takes place in specific contexts, and then by examining these background contexts. Much research has taken an ecological perspective on human development in their cultural contexts, such as ecological systems theory of Bronfenbrenner [2], the developmental niche approach of Super and Harkness [3] and the psychocultural model of Whiting [4]. More recently the role of features of the habitat of a population in their collective societal development have been proposed by Van de Vliert [5,6] who argues that

that culture is niche construction by populations, and by Welzel [7] who has advanced the cool water hypothesis to account for variations in societal development and shared values.

Bringing these ecological and cultural perspectives together, Berry [8,9] proposed an ecocultural framework (Figure 1) for understanding the origins, development and expression of human behaviour in context. This framework examines the roots of human cultural and psychological diversity by looking at two fundamental sources of influence (ecological and sociopolitical factors) and two features of human populations (cultural and biological adaptations to these factors). These group characteristics are transmitted to individuals by various "transmission variables" such as genetics, enculturation and acculturation. On the right are the behavioural consequences of these inputs and transmissions from them to individuals.

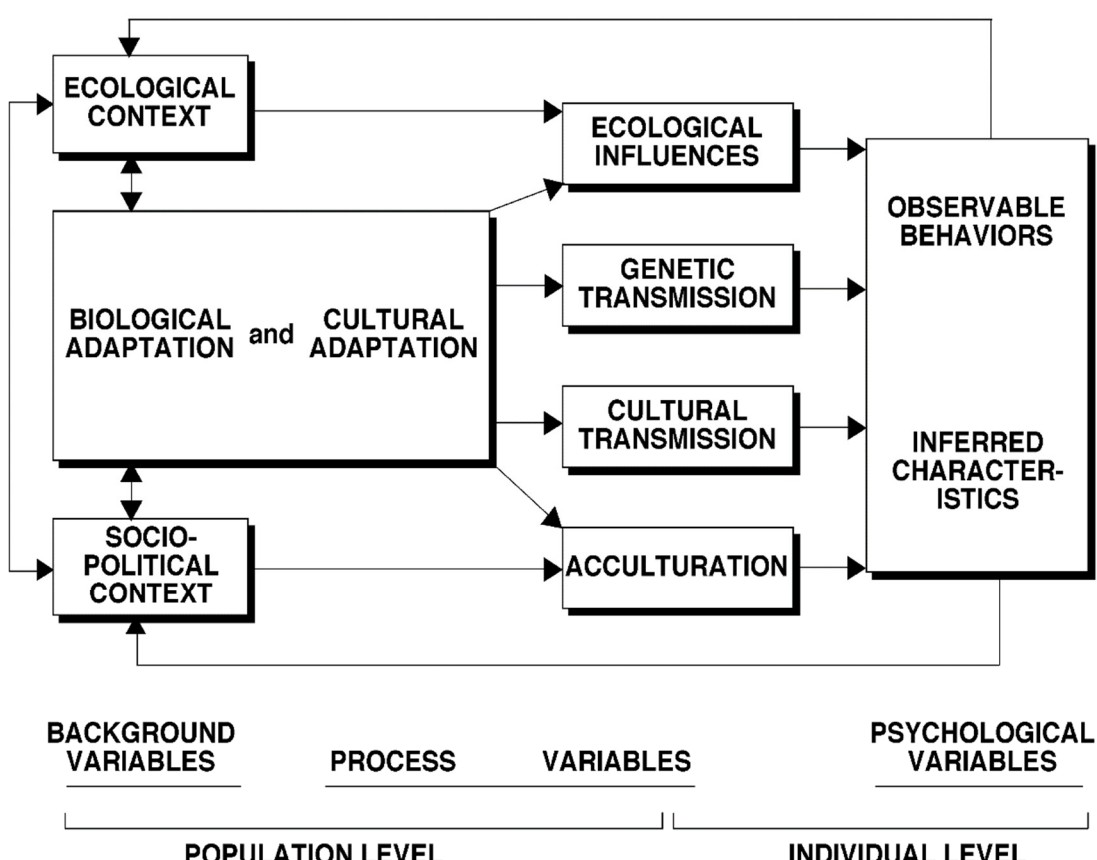

**Figure 1.** An Ecocultural Framework for examining Relationships Between Contexts, Cultural and Biological Adaptation, Cultural Transmission, Acculturation and Individual Adaptive Behaviour [9].

The ecological background (upper left of Figure 1) provides the physical contexts in which populations attempt to live; these populations adapt culturally and genetically to these features of their habitat over generations and share them with subsequent generations. The sociopolitical level (lower left) includes intercultural contacts and experiences that set the process of acculturation in motion. As a result of such contact. individuals have to adapt to more than one cultural context in which individual psychological phenomena can be viewed as attempts to deal simultaneously with two (sometimes inconsistent, sometimes conflicting) cultural contexts. These two inputs are conceptually and empirically related: contact takes place in ecological settings that are attractive to the colonisers or migrants [10], and acculturation outcomes from internal migration has been shown to be related to ecological factors in the society [11].

As a cultural institution, the family is an adaptation to both ecological and sociopolitical contexts. Large variations in family structures, practices, and values are known to vary

as adaptations to these ecological and sociopolitical contexts [12–15]. The family thus occupies a central place in the ecocultural approach, serving to link these background contexts to family structures and to individual behavioral development through the processes of enculturation and acculturation. The work of Kagitcibasi [16,17] is particularly relevant to understanding the role of these contexts in the adaptations of families and children to the original ecocultural and the new acculturation features of their lives.

In summary, the ecocultural framework considers the expression of human diversity (both cultural and psychological) to be a set of collective and individual adaptations to contexts. This framework assists us in the search for key features of both the long-term adaptation of cultures and families to their original habitats, and also to the newer contexts that are experienced following migration.

### 2.2. Acculturation and Adaptation

The study of the process of acculturation is central to describing and understanding how migrant families and individuals try to settle into and adapt to their new society. An early definition of *acculturation* by Redfield et al. [18] was a process of cultural change that follows cultural contact. Later, the concept of *psychological acculturation* was introduced by Graves [19] who noted that individuals as well as cultures also change psychologically following contact. Although original definition considered that first-hand contact was necessary for acculturation to happen, recently Ferguson et al. [20] have shown that acculturation can take place at a distance (termed *remote acculturation*) by way of media, without any direct contact between individuals.

All the cultural and psychological features that are brought by migrants to the acculturation arena, and those that are already present in the society of settlement, play a role in the eventual adaptation of families and individuals. There are usually differences between these two cultural populations in their values, beliefs and acculturation strategies, and between generations within the migrant families [21,22]. There are also differences between spouses within families, where there are often differing views about how to live in the new society [23]. All these differences may create challenges and conflicts within families.

Following the intercultural contacts that after migration, cultural communities, families and individuals go through the process of acculturation, eventually achieving various forms of adaptation [24]. The core meaning of the concept of acculturation refers to the process of cultural and psychological change in all groups and individuals that takes place as a result of contact between cultural groups and their individual members (as originally defined by Redfield, Linton, and Herskovits [18]). Such contact occurs for many reasons, such as colonization of Indigenous Peoples [25], ecological challenges [26], and political and social conflicts [27]. It continues after initial contact among diverse groups who are settled in culturally plural societies, where ethnocultural and indigenous communities maintain and change features of their heritage cultures over generations. It is important to note that acculturation takes place in all groups and all individuals that are in contact.

A framework to show the main features of this acculturation process and adaptation outcomes that flow from intercultural contacts is presented in Figure 2. (This is the sociopolitical input that was shown on the lower level of Figure 1).

The framework in Figure 2 links cultural (on the left) and psychological domains (on the right) of acculturation. It provides a map of those phenomena that need to be conceptualized and measured during acculturation research. To start, we need to understand the original features of the two or more cultural groups prior to their major contact. It is also important to understand the nature of their contact relationships, and the resulting cultural changes in the groups that emerge during the process of acculturation.

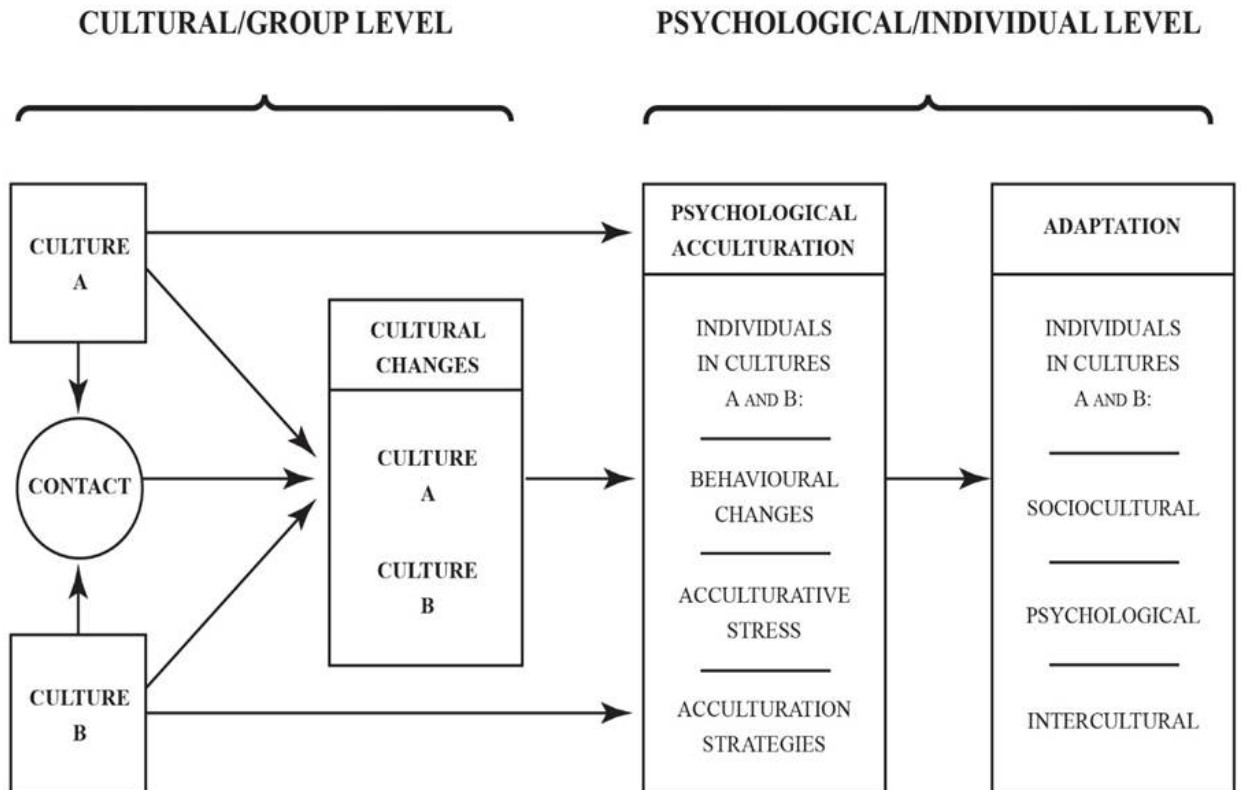

**Figure 2.** Framework for Examining Acculturation and Adaptation of Groups and Individuals [5].

The concept of *psychological acculturation* is shown in the middle of Figure 2 and refers to changes in individuals both in migrant families, and in the larger society, who are participants in a culture-contact situation. These changes can be a set of rather easily accomplished *behavioral shifts* (e.g., in ways of speaking, dressing, and eating), or more challenging (e.g., in values, and personality). When these changes are problematic, they produce the experience of *acculturative stress*, which is often manifested by uncertainty, anxiety, and depression.

The main reason for keeping the cultural and psychological levels distinct in Figure 2 is that not every individual or family enters into, participates in, or changes in the same way. There are vast differences in how people acculturate even among people who live in the same group or family. This variation in how people acculturate has led to the creation of the concept of *acculturation strategies* (see below). This concept refers to the different ways in which individuals and groups seek to engage the process of acculturation, usually resulting in different degrees of adaptation.

The concept of *adaptation* (on the right) refers to the longer term outcomes of the process of acculturation. Eventually acculturation results in some form of mutual accommodation between groups and among individuals. There are three kinds of adaptations to acculturation: they can be primarily internal and psychological (e.g., a sense of well-being or self-esteem, sometimes referred to as *feeling well*), sociocultural (e.g., competence in the activities of daily intercultural living; *doing well*), and intercultural (e.g., low levels of prejudice and a positive multicultural ideology; *relating well*).

Examining the processes of acculturation and adaptation of groups, families and individuals requires the examination of all these concepts (acculturation, acculturation strategies and adaptation) in order to understand the *what*, *how*, and *how well* of families and youth following their migration [10,24].

*2.3. Acculturation Strategies*

As noted above, not every group, family or individual seeks to engage the acculturation process in the same way. People live with and between two or more cultural groups, and may be oriented positively or negatively to them [28]. These *acculturation orientations* to the two cultures intersect to create four *acculturation strategies*. Both these concepts are used to refer to the various ways that people acculturate. Figure 3 shows these various ways for members of non-dominant groups on the left, and for the larges society on the right.

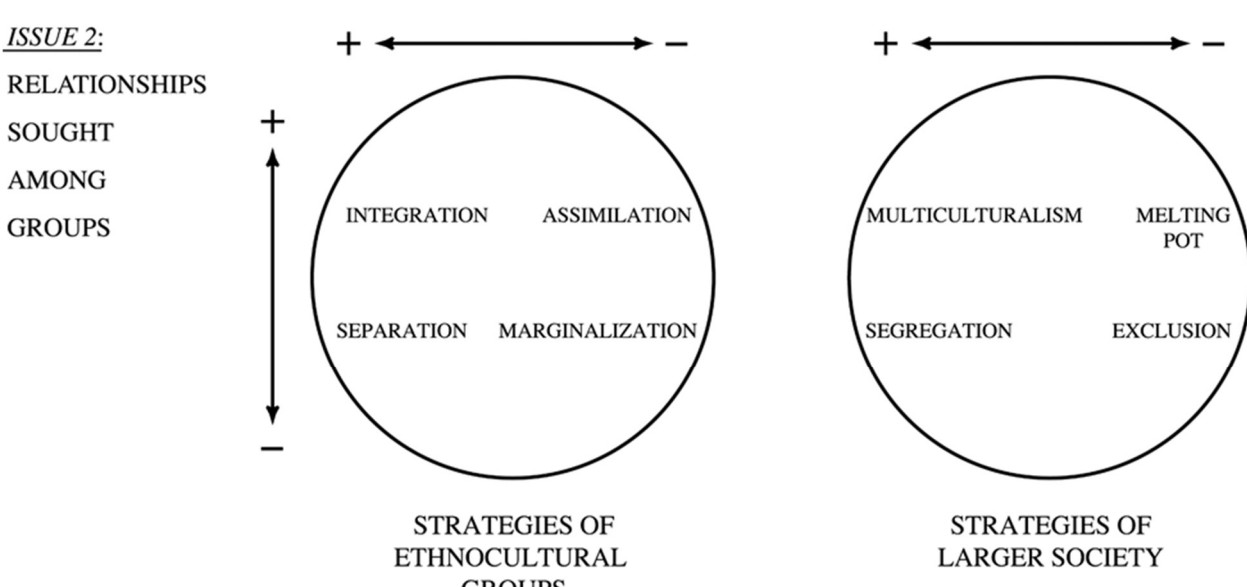

**Figure 3.** Framework for Examining Acculturation Strategies and Expectations in Ethnocultural Groups and the Larger Society [5].

The figure shows two acculturation *orientations*: a relative preference toward maintaining one's heritage culture and identity (along the top); and a relative preference for having contact with and participating in the larger society along with other ethnocultural groups (down the side). Four acculturation *strategies* have been derived by crossing these two acculturation orientations. Note that there is a fundamental distinction between acculturations *orientations* and acculturation *strategies*: orientations are toward two or more cultural groups; the four strategies derive from the intersection of these two orientations.

Groups in contact (whether non-dominant or dominant) usually have some notion about how they are attempting to engage the process of acculturation. Among ethnocultural groups (on the left), there are often goals that are articulated that may seek (or not) to maintain their heritage cultures, and to have contact with others outside their group. In the larger society (on the right), colonial or settlement policies and practices may seek to eliminate or perpetuate the cultures of migrants; or conversely they may or may not seek to have contact with migrants.

At the individual and family level, the goals of immigrants may vary within their family, for example on the basis of their educational or occupational background, and personal values. These variations in goals and motivations have led to them being considered to be acculturation *strategies*. They are more than just preferences or attitudes; they also have motivational qualities that promote the attainment of their goals.

Four acculturation strategies held by members of ethnocultural groups and individuals are named in the circle on the left of the figure; those held by members of the larger society

are in the circle on the right. Orientations to these two issues intersect to define four acculturation strategies. When members of ethnocultural groups do not wish to maintain their cultural identity and seek daily interaction with, and participate with other cultures in the larger society, the *assimilation* strategy is defined. When such individuals place a high value on holding on to their original culture, and at the same time wish to avoid interaction with others, then the *separation* alternative is defined. The *integration* strategy is defined when there is an interest in both maintaining one's original culture, while being in daily interactions with and participating along with other groups in the larger society. And when there is little possibility or interest in cultural maintenance, and little interest in or opportunity for having relations with others, then *marginalization* is defined.

The original definition of the process of acculturation [18] clearly established that all groups and individuals in contact would experience acculturation and change. The four terms used above described the acculturation strategies of non-dominant peoples. Different terms are needed to describe the strategies of the dominant larger society; these other terms are presented in the circle on the right side of Figure 3. Because they concern the ways that the larger society expects everyone to acculturate, they have been referred to as *acculturation expectations*. If the dominant group expects assimilation, this is termed the *melting pot*. When separation is enforced by the dominant group it is called *segregation*. Marginalization, when imposed by the dominant group, is called *exclusion*. Finally, for integration, when cultural diversity and equitable participation of all groups are widely accepted features of the society as a whole, it is called *multiculturalism*. The terms used for these expectations can be used to examine the attitudes and practices of members of the larger society and to identify the policies advocated by the larger society for dealing with migrants and ethnocultural groups.

A very important question is whether the acculturation strategies or expectations pursued have any relationship to the three forms of adaptation mentioned above. Research has shown that indeed those seeking the integration/multiculturalism way of acculturating achieve the best adaptations, while those who are marginalized/excluded have the poorest outcomes. Assimilation and separation strategies are typically associated with intermediate levels of adaptation. This relationships has been termed the *integration hypothesis* [29] and findings in support of it have been termed the *integration principle* [30].

*2.4. Acculturation and Cultural Transmission*

The ecocultural framework (Figure 1) presented various routes by which features of cultures are transmitted to the developing individual. In this section, we emphasise two forms of cultural transmission: *enculturation* and *acculturation*. These concepts are shown in Figure 4; they illustrate the ways in which a group can perpetuate its cultural and behavioral features among subsequent generations.

On the left of Figure 4, transmission by way of enculturation (within the original culture) may take place by three routes. First, enculturation from parents to their offspring is termed *vertical transmission,* since it involves the transmission by descent of cultural and behavioural characteristics down from the parental generation to the next within the family. The other forms of enculturation are *horizontal transmission* (from peers, such as in the contacts in the classroom or among gang members) and *oblique transmission* (from others of the parental generation in society, such as in clubs, schools and religious organisations).

Transmission by way of acculturation is shown on the right of Figure 4. Again, there are three forms of transmission, but they now arrive from another cultural group with which the group and individual are in contact. Parents are changed by their own experiences of acculturation (horizontally from the outside culture, at the top), leading to changes in vertical transmission from parents down to their children. Institutions (especially schools) in the new society can also change the developing individual by oblique transmission, without parental mediation. And of great importance is horizontal transmission from peers (in schools or clubs) who are members of the new larger society.

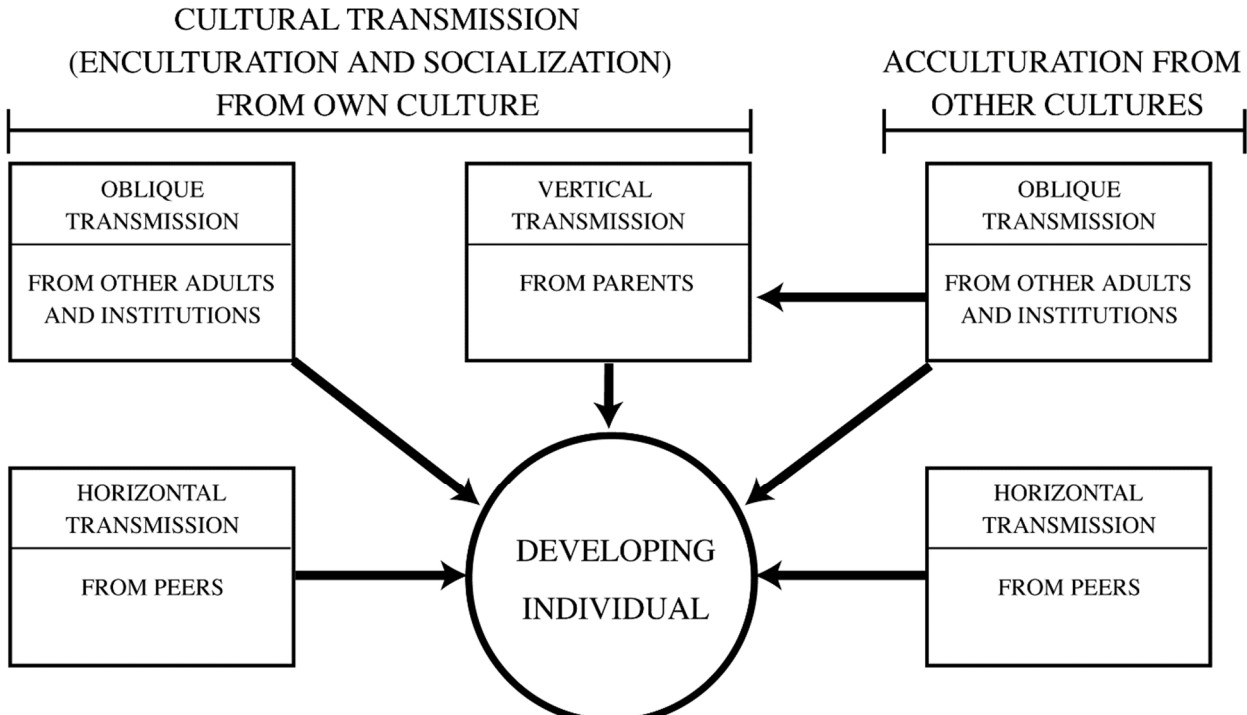

**Figure 4.** Framework for Examining Cultural Transmission (Enculturation and Acculturation) to Developing Individuals.

The ways in which features of both the original and new cultures become incorporated into families and their individual members can be observed and interpreted by the use of this cultural transmission framework. Features can be tracked, and the links can be identified. For example, there have been studies of the transmission of family relationship values (such as family obligations and adolescent autonomy) held by parents and their children, and by peers and other members of the larger society, to show the routes and extent of value transmission to the children. The similarities and correlations in behaviours, such as attitudes, values, identities, can be linked across all these groups. For example, are the values of a particular child related to those of their parents, their peers or with the shared values in the general population (called the *zeitg*eist)? Or in the case of migrant youth, are their values related to their heritage culture or to the values in the new society? One study [31] showed that these values can be traced to all these sources, including parents peers and other adults in both the original culture and in the acculturation arena.

**3. Results and Discussion: Some Empirical Examples**

To illustrate these frameworks, some empirical examples are now reviewed.

*3.1. Enculturation*

First, in relation to the ecocultural framework, evidence has been provided by Barry and his colleagues [32] to show that the processes of enculturation vary across cultures and are adapted to the ecology (especially their economic subsistence practices) of societies. They found that various domains of enculturation tended to form two clusters, termed "pressure toward compliance", and "pressure toward assertion". The importance of this research for family acculturation is that families migrating from different cultures around the world have long-established enculturation goals, and practices to achieve them. They bring these goals and practices to their society of settlement, and these may be consistent or at odds with those used in the new society. For example, the 'value of children' [17] and the family relationship values [33] discussed above are brought with the migrating families

to their new societies. These may differ sufficiently from those in the new setting to bring about acculturative stress and poor adaptation.

### 3.2. Family Structure

Beyond the ways that children are enculturated, these practices, and the family itself are known to be adapted to the ecological context [8,34]. The family occupies a central place in the ecocultural approach, serving to link background contexts to individual behavioral development in their respective habitats. Many empirical studies have demonstrated the existence of these relationships and have expanded the network of such relationships. For example, role differentiation (i.e., the number of specialized tasks that are distinguished within the society) and social stratification (i.e., the hierarchical arrangement among these roles, leading to variations in status) are important elements of these complex patterns. These variations in family structures are usually brought by immigrants to their new society, and continue to be used as ways to raise their children in the larger society.

Beyond the ecological context, the sociopolitical context (at lower level of the ecocultural framework) has played an important role in shaping both the cultural adaptation and transmission features of families. In particular, colonisation has brought about cultural changes that altered family arrangements and emphases in enculturation. They introduced new religions and forms of education, particularly formal schooling. Telemedia continue to promote change from outside by portraying alternative lifestyles, diet and consumer goods, as demonstrated by Ferguson and colleagues [20].

### 3.3. Families across Cultures

To illustrate the way in which the ecocultural approach (and its enculturation and acculturation components) may be used in the examination of families across cultures, we now review a project that incorporates all these features [13]. This project examined the relationships among ecocultural variables (both ecological and sociopolitical), economic practices, social structural variables, family roles, and their impact on some psychological variables.

The study examined these relationships in 27 nations around the world. Since much contemporary migration stems from the Third World, and settles in Europe and North America, it was important to identify the characteristics of families at both ends of the migration flow.

We created a number of variables related to family functioning. Among them are affluence, family roles and family relationship values.

#### 3.3.1. Affluence

We created a variable that we termed *affluence* based on the percentage of the population engaged in agriculture (derived from the ecological variable in the ecocultural model). It also includes sociopolitical influences from outside (formal education, which is derived from the sociopolitical variable in the ecocultural model).

#### 3.3.2. Family Roles

A number of roles were examined, including family traditions, kin relationships, hierarchy, housework, support of children, finances, and children helping parents with economic activities. These roles were asked for a number of family positions: father, mother, grandfather, grandmother, uncle/aunt, and children. Factor analysis produced evidence for three main roles for father and mother: expressive (e.g., providing emotional support to children, grandparents, and wife/husband); keeping the family united; keeping a pleasant environment; financial (e.g., contributing financially to the family, managing finances, supporting career of children); and childcare (e.g., taking children to school, playing with children, helping children with homework).

### 3.3.3. Family Relationship Values

Two family relationship values were found: *hierarchical roles* of father and mother; and *kin relationships.* Hierarchical values were negatively correlated with affluence and positively with percentage of the population engaged in agriculture. These hierarchical roles value the father as the patriarch of the family, who has the power and makes the important decisions, whereas the mother's role is to obey the father and raise the children.

We found that in societies low in affluence (high in agriculture) there were: closer family roles across the three generations than in high-affluence societies; closer expressive and child-duty roles for the mother, and grandmother; higher financial and instrumental roles of the father and grandfather; and higher family values placed on hierarchy and kin relationships, as well as greater similarity in these family values across generations.

Affluence thus appears to be the primary driver of many of these relationships. This study highlights the fundamental role of the affluence of a society in the patterning of social and psychological characteristics of the population and in families. Because affluence is a complex variable, combining ecological (e.g., agriculture) and sociopolitical (e.g., education) factors, we consider these consistent findings as support for the value of using the ecocultural approach to the study of family. Knowing the cultural characteristics that migrating families and individuals bring with them provides a basis for improving their acculturation and adaptation following migration.

### 3.4. Immigrant Youth

A number of studies have examined the process of acculturation and adaptation of immigrant youth [21,35]. To illustrate this work, we present the main findings from the International Study of Ethnocultural Youth (ICSEY) [32,36].

The project included immigrant youth who came from 26 different cultural backgrounds and lived in 13 countries. In each country we sampled both national and immigrant youth. We assessed the following variables: Acculturation Strategies; Cultural Identity; Language Proficiency and Language Use:;Ethnic and National Peer Contacts; Family Relationship Values; Perceived Discrimination; and Psychological Adaptation and Sociocultural Adaptation:

Cluster analysis with all the variables associated with the acculturation process yielded four clusters or profiles: integration (36%), separation (23%), assimilation (19%) and marginalization, (called diffuse, 22%). Youth in the separation profile showed a clear orientation toward their own ethnic group and showed little involvement with the larger society. The assimilation profile showed a strong orientation toward the new society, and a lack of retention of their ethnic culture and identity. The integration profile had those who indicated a relatively high involvement in both their heritage ethnic and national cultures. We termed the fourth profile *diffuse* (resembling marginalisation) for youth who were uncertain about their place in society, perhaps wanting to be part of the larger society, but lacking the skills and ability to make such contacts.

The profiles were analysed for differences in relation to length of residence to reveal differences over time since immigration. The profiles showed a clear pattern of differences across the three length-of-residence categories. The integration and national profiles were more frequent among those with longer residence. In contrast, the diffuse profile was much less frequent in those with longer residence. With respect to the experience of discrimination, these encounters were negatively related to adolescents' involvement in the larger society, being less frequent in the national and integration profiles, than in the separation and marginalisation profiles.

Of particular importance to understanding the acculturation and adaptation of migrant youth is the issue of whether immigrant youth in the different acculturation profiles adapt psychologically and socioculturally to different degrees. The answer is 'yes'. Immigrant youth in the integration profile have both adaptation scores that are above the grand mean of the sample, while those with the diffuse profile are below the grand mean. Being in the ethnic profile contributed positively to psychological adaptation (but not to sociocultural

adaptation), and a national orientation was related positively to sociocultural adaptation (but not to psychological adaptation).

This conclusion is strongly supported by the analysis of Schmitz and Schmitz [37] who examined the differences across acculturating individuals who used the four acculturation strategies for many psychological characteristics. In general, those who seek and achieve integration have better psychological adaptation, life satisfaction, self-esteem, and they have fewer mental health problems, lower levels of stress, anxiety and depression. They also found differences in personality, with integration being associated with lower levels of neuroticism, psychoticism, aggression and anger, and higher levels of extraversion, openness and agreeableness, This pattern of psychological characteristics clearly demonstrates the advantages of using the integration strategy during the acculturation process.

In sum, this study has shown that immigrant youth vary in their preferred ways of acculturating, and that these different acculturation strategies impact their psychological and sociocultural adaptation. With respect to the acculturation and strategies frameworks, the study also showed that many psychological characteristics are carried forward from their heritage cultures and are modified over time following migration.

## 4. Conclusions and Implications

The ecocultural approach, on which this paper is based, examines individuals families in cultural and intercultural contexts It has allowed for the identification of background ecological and sociopolitical factors that may influence the social, cultural and family characteristics of a society and the development of individual behaviors that are adaptive to them. Some similarities and differences in families have been connected using this approach, in which the family has been placed center stage in these arrangements, being both adaptive to context and serving as the main vehicle for cultural transmission, and as the basis for individual development. Overall, the ecocultural approach has served as a theory-based way to structure relationships among a complex set of variables when attempting to understand linkages between cultural and family contexts and variations in cultural transmission and individual behaviors and adaptations following migration.

The acculturation frameworks employed in this paper have allowed for the exploration of many variables that related to intercultural contact, cultural and psychological change and to the eventual adaptation of families and their individual members following migration.

The implications of knowing about and using the three core concepts (ecological adaptation to habitat, acculturation experiences, acculturation strategies, and individual adaptation) are widespread. In culturally diverse societies, these concepts, research frameworks, and empirical findings show how, and how well, people adapt in different ways in thie original habitats, and then further adapt as they engage each other across cultural boundaries.

In particular, knowing the benefits of pursuing and achieving integration (as understood here as the joint involvement in both cultural contexts) are far-reaching for those engaged in intercultural living. Of greatest importance is avoiding marginalisation, as well as the experience of discrimination, to achieve inclusion is an equally important finding.

Individuals and families who engage in the acculturation process can be informed by this research, and be shown the advantages of the integration acculturation strategy. Members of the larger society can likewise be provided with evidence that supporting the multicultural approach to managing intercultural relations is associated with more positive outcomes for everyone in the larger society. This is the case for both those who have immigrated to the society, and those who are already settled. All groups and individuals can benefit from being made aware of the advantages of integration and multiculturalism.

**Funding:** This research received no external funding.

**Conflicts of Interest:** The author declares no conflict of interest.

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
