# Peer review of "Family and Youth Development: Some Concepts and Findings Linked to The Ecocultural and Acculturation Modelsâ€"

_societies, doi:10.3390/soc12060181_

Round 1
Reviewer 1 Report
This is a review for the conceptual paper “Family and Youth Acculturation: Some Concepts and Findings”. The manuscript provides an overview of how we can look at family in an acculturation scenario, and draws on specific models (i.e., the ecocultural framework) to provide readers with a conceptual overview in how to understand acculturation in general (p.2-6). I find this informative for readers not yet familiar with the reasoning behind this model, but also think that the application and perspective at the end of the paper would benefit from possibly restructuring and connecting with more literature.
In the part specifically on cultural transmission, a further model from a relatively conceptual level is introduced. The model is clearly useful, but I think it would be a useful idea to orient readers to other general ideas specifically from a developmental perspective (particularly Bronfenbrenner, and proximal/distal considerations). I think there is a risk that readers otherwise take away that these are the main models we should be looking at. This does of course not mean that the considerations are not relevant after p.6, quite the contrary.
I could imagine that readers would benefit from specific references to other concepts that would help them orient towards the literature and open doors. An example would be that climate (Van de Vliert, 2007, 2016) also is increasingly understood as a factor in explaining differences in acculturation experiences (English et al., 2019). I believe this could be helpful for readers to understand the ecological background variables explained on page 2. Specifically for the notion of family the work by Cigdem Kagitcibasi is missing from the overview. A developmental perspective on how cultural differences come about that includes economic/material considerations like her Values of Children study is well-suited to explain to readers how these are linked in terms of cultural adaptation. Turkey’s intranational variation (with largely traditional rural context but also highly modern metropolitan areas) is a good showcase for that (Kagitcibasi, 2002; Kagitcibasi & Ataca, 2005).
I appreciate that section #6 provides some examples, but these are not connected with the previous conceptual models (or highlight other models), and seem very much like a list. The sections after p. 7 are also very different in nature, and it could be useful to assess whether removing enculturation from this section would help balance it. The Barry et al., 1959 reference could still be a line in the discussion.
Section 6.2 is labeled family structure, and seems to attend to structural elements in the first paragraph (focused on James Georgas and John Berry’s work), and the second paragraph focuses on cultural transmission (not so much structure). I think this opens up the possibility to talk about cultural transmission in more detail, with regard to acculturation, enculturation, and both in its remote variant (the work by Ferguson is briefly mentioned, and makes for a compelling empirical showcase).
Section 6.3 is a clear summary of the big comparative study by Georgas et al.. In the spirit of opening doors, some of the introductory text could be condensed in favor of a clearer summary of outcomes. The ICSEY data in section 6.4 could possibly benefiting from providing fewer details (it does not seem necessary to list variables), this would leave more room for connecting with other literature.
I think a longer discussion or perspective section would be useful to understand how we can use prior findings (e.g., the ones reported) to address current challenges in cultural transmission. An application to specific problems is, in my view, missing. There is much research on youth acculturation, specifically the immigrant paradox (Garcia Coll), differences between groups, and an apparently increased attention to identity processes (including intersectionality concerns).
From a more sociological perspective I think it could be clarified more whether we need to understand family migration (i.e., the family moving together to a new context of residence) and stepwise family reunification are similar contexts.
Minor notes
- I think the title can be more informative, some concepts and findings is correct, but does not reveal much about the direction of the paper. It could focus on variation and challenges, for example. The conclusion, for example, specifies that the manuscript is based on the ecocultural approach – which could feature in the title, and provide a clearer red ribbon. On the other hand, not all empirical examples in section 6 completely align with this focus, and I think the reader would also benefit from learning about other perspectives.
- This also goes for the header 6 - Results and Discussion - Some Empirical Examples. I do not find that very informative. I think there can be good reasons why we should attend to enculturation as a process, family structure, comparative perspectives, and specifically immigrant youth
- Happy to see a dedication to James Georgas at the beginning of the paper
- Figure 6 is very blurry.
References
English, A. S., Kunst, J. R., & Sam, D. L. (2019). Climatic effects on the sociocultural and psychological adaptation of migrants within China: A longitudinal test of two competing perspectives. Asian Journal of Social Psychology, ajsp.12363. https://doi.org/10.1111/ajsp.12363
Kagitcibasi, C. (2002). A Model of Family Change in Cultural Context. Online Readings in Psychology and Culture, 6(3), 1–11. https://doi.org/10.9707/2307-0919.1059
Kagitcibasi, C., & Ataca, B. (2005). Value of Children and Family Change: A Three-Decade Portrait From Turkey. Applied Psychology, 54(3), 317–337. https://doi.org/10.1111/j.1464-0597.2005.00213.x
Van de Vliert, E. (2007). Climates Create Cultures. Social and Personality Psychology Compass, 1, 53–67. https://doi.org/10.1111/j.1751-9004.2007.00003.x
Van de Vliert, E. (2016). Human Cultures as Niche Constructions Within the Solar System. Journal of Cross-Cultural Psychology, 47(1), 21–27. https://doi.org/10.1177/0022022115615963
Reviewer 2 Report
The manuscript is about a really interesting topic, and it describes a conceptual theorethical framework to better understand intercultural adaptation. However, I think it does not provide new insights into new ways of looking at existing knowledge and concepts. It reports existing conceptual models and the results of some previous studies, but I do not think it adds something to the current literature.
Author Response
Dear reviewer,
The manuscript had been revised in consideration of comments from both reviewers, please see the revised version with tracked changes.
Please note that a point-by-point response to your comments were not provided as your comments from the first round are relatively brief, so please check whether the recent revisions have addressed your concerns in the first round of review.
With best regards,
The Editorial Office
Round 2
Reviewer 1 Report
The authors have revised the manuscript and added several passages, addressing the majority of my previous comments.